



# Decadal Transition of Summertime PM₂.₅-O₃ Coupling and Secondary Organic Aerosol Dominance in Northwest China

Wei Zhou[1], Liu Yang[1,2], Siqi Zeng[1,2], Yunping Kan[1,2], Lirong Yang[3], Weihong Zhang[3], Weijie Wang[3], Zijun Zhang[1,2], Yan Li[1], Weiqi Xu[1], Yucheng Gu[3], Yaozong Wang[3], Zhengyan Zuo[3], Jie Li[1], Zifa Wang[1,2], Yele Sun[1,2]

[1]State Key Laboratory of Atmospheric Environment and Extreme Meteorology, Institute of Atmospheric Physics, Chinese Academy of Sciences, Beijing, 100029, China

[2]College of Earth and Planetary Sciences, University of Chinese Academy of Sciences, Beijing, 100049, China

[3]Ningxia Hui Autonomous Region Ecological Environment Monitoring Center, Yinchuan, 750000, China

*Correspondence to*: Wei Zhou (zhouwei215@mails.ucas.ac.cn) and Jie Li (lijie8074@mail.iap.ac.cn)

**Abstract.** The Yinchuan metropolitan area in northwest China, situated between the Tengger and Ulan Buh Deserts, is influenced by both natural dust and anthropogenic emissions. However, the evolution of fine particulate matter (PM₂.₅) and its interaction with ozone (O₃) under the region's arid climate remain poorly understood. This study integrates decadal observations (2015-2025) with in-situ measurements using an Aerosol Chemical Speciation Monitor and a Vocus Proton Transfer Reaction Mass Spectrometry during summer 2025 to elucidate the changing PM₂.₅-O₃ relationship and sources of organic aerosols. A pronounced shift was identified: Phase I (2015–2018) featured a rapid decline in PM₂.₅ accompanied by a sharp O₃ increase, while Phase II (2019–2025) exhibited stabilized PM₂.₅ and plateaued O₃, indicating reduced O₃ sensitivity to particulate controls. The average non-refractory PM₂.₅ concentration (16.8 µg m⁻³) was significantly lower than in eastern Chinese megacities, with organics accounting for ~60 %. Positive matrix factorization resolved three organic aerosol factors, revealing dominant secondary organic aerosols (SOA, ~74%) derived from prolonged photochemical aging. Volatile organic compound analysis showed that anthropogenic and biogenic precursors, including urban terpenes and aromatic oxidation products jointly contributed to SOA formation. Back-trajectory and potential source analyses indicated that Yinchuan's summer air masses were mainly locally recirculated, with limited influence from long-range transport. These results demonstrate a regional transition toward SOA-dominated fine particles and decoupled PM₂.₅-O₃ dynamics under cleaner conditions, highlighting the need for integrated VOC and oxidant controls to mitigate co-occurring O₃ and PM₂.₅ pollution in arid northwest China.



## 1    Introduction

Fine particulate matter (PM$_{2.5}$) has a significant influence on human health, the environment and climate (Brauer et al., 2024),
which has been included in the Chinese air quality standards for the first time in 2012 (Zhang et al., 2012). Since then, concerns about PM$_{2.5}$ has been increasingly growing across various sectors of society and researchers. Over the past decade, numerous initiatives aimed at declining particulate pollution in China have been implemented nationwide, resulting in substantial decreases in particulate concentrations, from hundreds to dozens of µg m$^{-3}$ in the past decade (Zhang et al., 2019a;Geng et al., 2024). Despite this, sporadic pollution events still occurred in China, particularly in underdeveloped areas with heavy industrial
emissions (An et al., 2019). In addition to improvements in air quality, advancements in mass spectrometry have greatly enhanced our understanding of the compositions of aerosols (Zhou et al., 2020;Sun et al., 2025), particularly regarding the formation mechanisms of organic aerosols (OA). For example, anthropogenic secondary OA mainly originated from the oxidation of aromatic volatile organic compounds (VOCs) in urban areas, while the photochemical oxidation of biogenic isoprene during the daytime and the oxidation of monoterpenes and sesquiterpenes by nighttime NO$_3$ radicals are the primary
pathways for the production of biogenic secondary OA (Mehra et al., 2021).

Extensive studies have been conducted on the aerosol compositions and sources of PM$_{2.5}$ across China. These studies have demonstrated that organic aerosols are the most important component of atmospheric aerosols, with their contribution ranging from 30-40% in urban areas to as high as 80-90% in rural and suburban regions (Zhang et al., 2007;Zhou et al., 2020). This indicates that OA has become a major factor in the formation of severe pollution, particularly in underdeveloped regions.
As emission control measures continue to be implemented across China, gaseous precursors such as SO$_2$ and NOx have experienced sharp declines, resulting in a significant decrease in the inorganic components of PM$_{2.5}$, e.g., sulfate. In contrast, the contribution of organic aerosols, particularly secondary organic aerosols, has been increasing (Chen et al., 2024), although achieving a persistent decrease in organic aerosols mass concentrations remain a challenge. Specifically, different types of organic aerosols showed varied responses to long-term influences from anthropogenic emission and regional climate change
(Zhang et al., 2025). For instance, primary OA, isoprene-derived SOA and monoterpene SOA exhibited distinct variations as corresponding to changes in NOx and sulfate levels. These results suggest the complex sources of organic aerosols in the context of relatively lower pollution levels in China. A very recent study highlighted the pollution characteristics over China, revealing the complex interactions among various atmospheric environmental factors contribute to the unique formation mechanisms of SOA. This underscores the importance of interaction among multiple pollutants, for example, multiphase
reactions, and multi-generation chemical transformation process (Huang et al., 2025).

Previous field campaigns utilizing aerosol mass spectrometry predominantly concentrated in three metropolitan clusters in China, e.g., the North China Plain (NCP), Yangtze River Delta (YRD), and Pearl River Delta (PRD)(Zhou et al., 2020). In recent years, several measurements have also been carried out in the Fenwei Plain within the Yellow River Basin, specifically in Yuncheng, Shanxi province (Li et al., 2022;Wang et al., 2024a) and Baoji, Shaanxi province (Li et al., 2025). Moreover, an
increasing number of observations have been made in the Chengdu-Chongqing Basin in southwest China (Bao et al., 2023).



Their results revealed significant differences in the sources and compositions of organic aerosols across China, attributed to the influences of meteorological conditions, topographical features and local industries. For instance, the relatively high humidity in southwest China facilitated the formation of SOA, which contrasts sharply with the much drier climate of northwest China. Compared to these major regions, research focusing on northwestern China is still insufficient. Xu et al. (2014) conducted the first aerosol mass spectrometry measurements in Lanzhou, a major city in northwest China, finding that local traffic and coal combustion significantly contribute to OA levels in summertime, while the non-fossil sources such as biomass burning and cooking activities could accounted for 55% of OA in winter (Xu et al., 2016a). Despite these findings, our understanding of organic aerosol characteristics in northwestern China remains limited. This is particularly relevant given that Lanzhou is situated in a narrow valley between two mountain ranges, and its unique topographical features mean that it does not necessarily represent the characteristics of most areas in northwest China.

With China's implementation of the "go west" policy, understanding $PM_{2.5}$ levels and their sources in the northwest China has become a critical factor in policymaking for region development. The Yinchuan metropolitan area severs as one of the major city clusters in northwest China, located adjacent to the Tengger Desert and Ulan Buh Desert. This area is characterized by heavy industry and mining as its dominant sectors, resulting in a complex air pollution landscape derived from both natural and anthropogenic sources. The unique geographical features and proximity to dust source regions may lead to significantly different characteristics of $PM_{2.5}$ pollution compared to other areas in China. However, the long-term pollution characteristics of $PM_{2.5}$ and the increasing concerns regarding $O_3$ have been less thoroughly investigated. In this study, comprehensive measurements of $PM_{2.5}$ compositions and VOCs are conducted during the summer in Yinchuan, the capital city of Yinchuan metropolitan area. The research focused on the coupling relationships between $PM_{2.5}$ and $O_3$ over the past decade, examining the chemical compositions of $PM_{2.5}$ and their responses to changes in meteorological conditions. The source apportionment of organic aerosols and typical emissions of VOCs in Yinchuan are researched.

## 2    Experimental methods

### 2.1  Sampling sites and measurements

The summertime campaign was conducted at the Yinchuan Atmospheric Environmental Super Monitoring Station (38°28'24''N, 106°10'57''E; ALS: 1500 m), from June 6 to July 10, 2025. This station is located on the rooftop of Yinchuan No.6 Middle School, approximately 10 m above the ground level. The surrounding area mainly consists of residential neighborhoods and traffic roads, making it a typical urban site. Yinchuan, the capital of the Ningxia Hui Autonomous Region, is geographically situated in an arid and semi-arid climate zone. As shown in Fig. 1, Yinchuan is bordered by desert and sandy terrain to the west (Tengger Desert), north (Ulan Buh Desert) and east (Mu Us Sand Land), which contributes to the city's susceptibility to dust storms, particularly during the spring and early summer seasons. Note that the Helan mountains (3556 m) are located to the west of Yinchuan city, approximately 30 km to our sampling site, which could partially mitigate the erosion of sand storms to some extent. In addition, the Yellow River flows from the southwest to the northeast across the Ningxia Autonomous Region, supplying abundant irrigation water and fostering relatively lush vegetation in Yinchuan.



A Time-of-Flight Aerosol Chemical Speciation Monitor (hereafter ToF-ACSM, Aerodyne Research Inc.) was deployed

for the real-time online measurement of non-refractory $PM_{2.5}$ (NR-$PM_{2.5}$) during this campaign, including organics (Org), sulfate ($SO_4$), nitrate ($NO_3$), ammonium ($NH_4$), and chloride (Chl), with a time resolution of 2 min (Fröhlich et al., 2015). A $PM_{2.5}$ cyclone was installed at the front of the sampling line to remove coarse particles larger than 2.5 µm, operating at a total sampling flow rate of 5 L min$^{-1}$. The sampled air was passed through a 1.2 m Nafion dryer for drying, with a dried subflow of 0.1 L min$^{-1}$ directed into the ToF-ACSM for detection. Details of the ToF-ACSM and instrument setups has been given by

Zhou et al. (2022). Note that, due to evaporation temperature limitations, refractory components such as black carbon and dust cannot be detected by the ToF-ACSM, which typically accounted for a non-negligible fraction in northwest China (Xu et al., 2014). In addition to the ToF-ACSM, collocated measurements at the supersite station included elemental carbon (EC) determined from a Sunset OC/EC analyzer, and 27 crustal elementals measured using X-ray fluorescence spectrometry (XRF), which together served as the refractory components of $PM_{2.5}$. The NR-$PM_{2.5}$ measured by the ToF-ACSM, along with EC and

the crustal elements, collectively contributed to the mass closure of total $PM_{2.5}$. The Al, Si, Ca, Ti and Fe were calculated as soil, while the remaining elementals were classified as metals within the total $PM_{2.5}$ (Xu et al., 2021).

Simultaneously, VOCs were measured at the same site using a Vocus Scott Proton transfer reaction time-of-flight mass spectrometry (hereafter Vocus PTR-MS; Tofwerk, Switzerland), with a mass resolution of ~4000 and a time resolution of ~1 s. Detailed information about the Vocus PTR-MS can be found in Zhang et al. (2024). The instrument was daily calibrated for

25 VOCs using customized standard gas mixtures, including benzene, toluene, isoprene etc. For species not directly calibrated, their sensitivities were estimated based on a linear fitting relationship between the sensitivities of calibrated species and their proton transfer reaction rate constants (Cappellin et al., 2012). Additionally, hourly meteorological parameters (wind speed, wind direction, relative humidity and temperature), as well as gaseous precursors (NOx, CO and $O_3$) were also obtained from the super monitoring station.

**2.2 Data analysis**

The ToF-ACSM data were processed using the standard software Tofware (v2.5.13) to obtain the mass concentrations of NR-$PM_{2.5}$ species and organic aerosol mass spectra.   A unity collection efficiency was employed for quantification due to the application of capture vaporizer in the ToF-ACSM (Xu et al., 2016b). Similarly, the Vocus PTR-MS data were also processed with Tofware v4.0.1 to obtain VOCs signals, and the corresponding sensitivities were applied to convert these signals into

mixing ratios of VOCs. To resolve the sources of OA, positive matrix factorization (PMF) was performed on the OA mass spectra. A three-factor solution was finally chosen in this study, comprising a primary OA (POA) factor and two secondary OA (SOA) factors.

The 72 h back trajectories of air masses arriving in Yinchuan were calculated hourly using the Hybrid Single Particle Lagrangian Integrated Trajectory (HYSPLIT, NOAA) model (Draxler and Hess, 1997) at a release height of 1500 m. The

potential source regions for five NR-$PM_{2.5}$ species and three OA factor were determined using Potential Source Contribution Function (PSCF) analysis (Polissar et al., 1999), based on the 72 h back trajectory data outputted from HYSPLIT. The 75[th]



percentiles in gridded cells were set as threshold for calculating the PSCF, while applying a weighting function to diminish the impact of low trajectory values.

## 3    Results and discussion

### 130    3.1    Co-existence of $PM_{2.5}$ and $O_3$ pollution in Yinchuan

Figure 2 illustrates the monthly and annual summertime average concentrations of $PM_{2.5}$ and $O_3$ in Yinchuan over the past decade. The annual average mass concentration of $PM_{2.5}$ decreased from 53.6 µg m$^{-3}$ in 2015 to 38.9 µg m$^{-3}$ in 2024, marking a reduction of 27% at an average rate of 1.96 µg m$^{-3}$ yr$^{-1}$. This trend aligns with the consistent decline in $PM_{2.5}$ levels across China, as a result of the implementation of emissions control measures (Zhang et al., 2019a), which also coincided with a

concurrent sharp decrease in $SO_2$. These results demonstrate effective nationwide $PM_{2.5}$ control measures, note only in the well-documented regions such as NCP, PRD and YRD, but also in northwestern China, including Yinchuan. However, while the most significant decline occurred in winter, Yinchuan continued to experience frequent heavy pollution incidents, with $PM_{2.5}$ levels peak above 200 µg m$^{-3}$. This highlight persistent challenges related to wintertime particulate pollution in the region, primarily associated with the enhanced coal combustion in heating seasons in northern China. It is noted that, despite

the overall downward trend in annual $PM_{2.5}$ in Yinchuan, a noteworthy rebound occurred after 2019. Specifically, from 2019 to 2024, the annual average $PM_{2.5}$ increased by 18%. In fact, since 2018, achieving further reductions in emissions has become increasingly difficult, complicating effects to control $PM_{2.5}$ levels despite ongoing reductions in precursor emissions (Lei et al., 2021). This struggle is not unique to Yinchuan; the NCP has also encountered challenges in achieving significant reductions during the second phase of pollution measures from 2018 to 2020. Consistently, the observed prevalence of SOA in fine particle

matter in Yinchuan confirmed that simply reducing primary emissions is inadequate for mitigating $PM_{2.5}$ pollution in this region.

From a seasonal perspective, average summertime $PM_{2.5}$ concentrations saw a sharp decline, dropping from 45.7 µg m$^{-3}$ in 2015 to 20 µg m$^{-3}$ in 2019. Interestingly, between 2019 and 2025, the average $PM_{2.5}$ levels remained relatively stable, fluctuating within a narrow range of 18-20 µg m$^{-3}$, except for the notable rebound in 2024, when concentrations reached 29.5

150    µg m$^{-3}$. The cause of this abrupt increase remains unclear, particularly given that the levels of gaseous precursors have remained consistent with previous years. Meteorological factors may offer the most plausible explanation. On one hand, the sustained $PM_{2.5}$ concentrations below 20 µg m$^{-3}$ after 2018 reflect the successful implementation of air pollution control measures in Ningxia, particularly in Yinchuan city, at least during the summer months, although $PM_{2.5}$ pollution continues to be a concern during the winter. On the other hand, in contrast to cities such as Beijing, where summertime $PM_{2.5}$ concentrations continued

to decline annually after 2018 (by 50% between 2018 and 2022)(Li et al., 2023a), Yinchuan exhibited minimal variation in $PM_{2.5}$ levels over the last seven years. Overall, our findings highlight the success of particulate pollution controlling during summer in Yinchuan for the past decade, more importantly, emphasize the greater difficulties in further $PM_{2.5}$ reductions compared to other regions.

Contrary to the significant decrease in $PM_{2.5}$ concentrations between 2015 and 2025, the annual average $O_3$ concentrations



increased at a rate of 0.96 µg m$^{-3}$ yr$^{-1}$ in Yinchuan. This is consistent with the overwhelming growing O$_3$ levels during the past decade from other regions, such as YRD and PRD, both in summer and winter seasons(Li et al., 2019b;Li et al., 2021a;Zhou et al., 2021). Specifically, the summertime O$_3$ in Yinchuan gradually rose from 80 µg m$^{-3}$ in 2015 to roughly stabilize at 105 µg m$^{-3}$ over the past three years. Indeed, it is important to that the relatively constant O$_3$ levels likely suggest that ozone pollution has been largely controlled in recent years in Yinchuan. Overall, our results reveal the uniformly increasing merging

O$_3$ pollution issues in Yinchuan, coinciding with decreasing PM$_{2.5}$ levels. This trend is consistent with Wang et al. (2020) reporting the contrasting trends between PM$_{2.5}$ and O$_3$ in China during 2013-2017. In fact, recent studies have shown that, due to the scavenging effects of particulate matter on HO$_2$ radicals and NOx, O$_3$ concentrations have risen despite reductions in PM$_{2.5}$ (Li et al., 2019a). Seasonally, this increase was more pronounced during summer and autumn, with an average annual rise of 2 µg m$^{-3}$ per year, while the increase in winter was modest at only 0.5 µg m$^{-3}$ yr$^{-1}$. However, Li et al. (2021a) found the

ozone pollution in NCP could spread into the late-winter haze season, much different from those in Yinchuan. Further analysis is needed regarding the coupling mechanism between PM$_{2.5}$ and O$_3$ production across different seasons, as well as the long-term changes in their sensitivities to precursor emissions, e.g., VOCs.

Notably, the rate of increase in O$_3$ in Yinchuan began to slow down after 2018 compared to preceding years, with only an 8% increase during the post-2018 period, in contrast to a 41% increase before that year. These findings highlight that although

the second-phase pollution control measures implemented after 2018 may not have successfully mitigated PM$_{2.5}$ pollution in Yinchuan, they appear to have had some effectiveness in controlling O$_3$ production. In summary, there has been a significant shift in the relationship between summertime PM$_{2.5}$ and O$_3$ in Yinchuan over the past decade. Phase I, from 2015 to 2018, was characterized by a sharp decline in PM$_{2.5}$ levels accompanied by a concurrent significant rise in O$_3$, primarily driven by substantial reductions in gaseous precursors. In contrast, Phase II, spanning from 2019 to 2025, was marked by a gradual

increase in O$_3$ levels while PM$_{2.5}$ concentrations remained stagnant, correlating with much slower reductions in precursor emissions, e.g., VOCs, NH$_3$ and NOx. The different trajectories of O$_3$ in response to changes in PM$_{2.5}$ between Phase I and Phase II were associated with complex factors, including the influence of precursors and OH radicals. A very recent study has also revealed similar trend of O$_3$ levels in Beijing as a response to national-wide emission controls during 2005-2020, characterized by an initial rapid increase followed by a gradual decrease, attributed to changes in the atmospheric oxidation

capacity (Wang et al., 2024b). This is consistent with Geng et al. (2024) showing that the "Ten Measures for Air Pollution Prevention and Control" achieved significant emission reductions in PM$_{2.5}$ in China during 2013-2017, however, the potential for further reduction after 2017 has become limited based on emission inventories and model simulations. Therefore, the timing of controlling VOCs and NOx is crucial keys for the coordinated reductions of both O$_3$ and PM$_{2.5}$ levels in Yinchuan, and even the synergistic tripe controls of O$_3$, PM$_{25}$ and CO$_2$ in the future (Liu et al., 2025).

We further explored the complex non-linear relationships between summertime PM$_{2.5}$ and O$_3$ in Yinchuan using exponential fitting methods similar to those proposed by Zhang et al. (2022). As shown in Fig. 3, our findings revealed that from 2015 to 2018, changes in the MDA8 O$_3$ exhibited a notable parabolic distribution trend correlated with PM$_{2.5}$, with a peak occurring at 45 µg m$^{-3}$. During this period, as PM$_{2.5}$ concentrations fell on the right side of the curve, they corresponded



perfectly to the sharp reductions in PM$_{2.5}$ levels seen during Phase I, leading to significant increases in MDA8 O$_3$ levels when summertime PM$_{2.5}$ exceeded 45 µg m$^{-3}$. However, from 2019 to 2025, variations of MDA8 O$_3$ were minimal in response to changes in PM$_{2.5}$, indicating that the continued decline in PM$_{2.5}$ did not significantly affect O$_3$ concentrations in this context. These findings suggest a substantial transition from Phase I to Phase II regarding the PM$_{2.5}$-O$_3$ relationships, with O$_3$ becoming less sensitive to controls on PM$_{2.5}$ (Liu et al., 2025). Overall, our results underscore significant differences in the complex interactions between PM$_{2.5}$ and O$_3$ during the two phases of summer in Yinchuan, primarily due to varying reduction percentages of gaseous precursors over the past decade. It is important to note that while the exponential fitting curve during Phase II was considerably flatter compared to Phase I, the timing of the inflection point advanced to correspond with lower PM$_{2.5}$ concentrations in Phase II. This suggests that efforts to control PM$_{2.5}$ are expected to more effectively manage O$_3$ levels over time. When comparing our results with those from other regions, the PRD and YRD exhibited monotonic variations between O$_3$ and PM$_{2.5}$, demonstrating the effectiveness of PM$_{2.5}$ controlling on O$_3$ in these areas (Zhang et al., 2022). In contrast, Beijing displayed a similar pattern to Phase I, initially showing an increasing trend followed by a decrease in summer (Zeng et al., 2024). These differences in the unique characteristics of PM$_{2.5}$-O$_3$ relationships in Yinchuan in recent years, likely attributable to the relatively low PM$_{2.5}$ concentrations in Yinchuan compared to other metropolitan clusters. The measurements of aerosol compositions and VOCs components will aid in further investigating the causes of the double high PM$_{2.5}$ and O$_3$ pollution in Yinchuan.

## 3.2 Chemical compositions of fine particles in Yinchuan

Figure 4 displays the meteorological parameters, gaseous precursors, and NR-PM$_{2.5}$ species measured during the campaign. In Yinchuan, summertime temperatures were generally below 35$^{o}$C, with an average of 26.4±4.5$^{o}$C. Surface winds were mild, predominantly coming from the north and northeast, with wind speeds ranging from 0 to 3 m s$^{-1}$. With regards to meteorological influences, relatively higher NR-PM$_{2.5}$ mass loadings were often associated with northerly wind. This correlation is likely due to the transport of pollutants from the northern part of the Ningxia Hui Autonomous Region, particularly from Shizuishan City, which is known for its significant industrial emissions. Although Yinchuan is located in an arid and semi-arid region, RH varied widely from 9% to 98%, averaged by 44±20%. The observed moderate RH levels were primarily influenced by frequent rain events during the sampling period, particularly in the latter phase of the campaign after 22 June. For example, significant precipitation events occurred on June 27 and 29, which may have contributed to the scavenging of particulate matter to some extent.

The concentrations of NR-PM$_{2.5}$ varied dynamically, with total mass ranging from 1.3 to 68 µg m$^{-3}$ and an average of 16.8±9.2 µg m$^{-3}$. Throughout the campaign, only 4% of the time of when hourly NR-PM$_{2.5}$ mass concentrations exceeded 35 µg m$^{-3}$, further suggesting that air quality in Yinchuan generally meets the National Air Quality Grade I Standard. In comparison to other summertime measurements insight from ToF-ACSM or AMS studies in China (Fig.5), the averaged NR-PM$_{2.5}$ in Yinchuan was significantly lower than those reported in cities within the North China Plain (NCP) and Pearl River Delta (PRD), e.g., Xingtai (Zhang et al., 2018), Handan (Li et al., 2018a), Nanjing and Hangzhou (Li et al., 2018b). Note that,



it was also lower than that reported for Beijing (18.1 µg m$^{-3}$) during the summer of 2022 by Zeng et al. (2024). Although these comparative observations were conducted prior to 2025, specifically mostly between 2013 and 2020, it is possible that our comparisons may exaggerate pollution levels in other cities, given the ongoing improvements in nationwide air quality in recent years. However, aside from a minor rebound in the summer of 2024, fine particulate concentrations in Yinchuan have remained largely constant since 2019. Therefore, the observations from 2025 are representative of overall PM$_{2.5}$ conditions in Yinchuan from 2019 to 2025, reinforcing the validity of the comparative analysis. These results indicate relatively better air quality in Yinchuan during summer period than major metropolitan cluster cities, even though it is still approximately double that of background locations (Zhang et al., 2019b). Most importantly, NR-PM$_{2.5}$ levels in Yinchuan were lower than those in nearby northwestern cities such as Lanzhou (Xu et al., 2014). This difference can likely be owing to the fewer industrial facilities in Yinchuan, as well as its more favorable atmospheric dispersion conditions, given that Lanzhou is situated in a valley surrounded by high mountains.

On average, the NR-PM$_{2.5}$ concentrations measured by the ToF-ACSM accounted for 71% of the total PM$_{2.5}$ mass (=NR-PM$_{2.5}$+EC+Soil+Metals), indicating that the ToF-ACSM effectively captured the majority of fine particulate matter components in Yinchuan. Among the refractory species, soil constituted 26% of PM$_{2.5}$, while metals contributed 1%, primarily due to calcium originating from suspended dust and construction activities. In comparison to dust-free cities, the proportion of NR-PM$_{2.5}$ within total PM$_{2.5}$ was lower in this study, e.g., Fenhe Plain (Li et al., 2022) and central China Plain (Li et al., 2021b). However, this finding is consistent with the regional context, as northwestern China is prone to dust events. Although summertime is generally less susceptible to dust storms, nearby sand sources still contribute to elevated levels of suspended dust in Yinchuan, resulting in much higher presence of refractory species in this study than previous studies. Our results revealed significant differences in aerosol components measured by different instruments. While secondary inorganic aerosols (SIA), including sulfate, nitrate and ammonium, demonstrated reasonably strong correlations (R$^2$=0.51-0.67) between the ToF-ACSM and online ion chromatography measurements, the overall NR-PM$_{2.5}$ mass showed a weak correlation with PM$_{2.5}$ data from the supersite monitoring station. Given that the organics detected by the ToF-ACSM also correlated well with those measured by the Sunset OC/EC analyzer, the discrepancy between NR-PM$_{2.5}$ and total PM$_{2.5}$ can be attributed to the totally different sources of SIA and water-insoluble dust materials, with the latter considerably contributing to the overall mass. Although traditional ion chromatography and OC/EC analysis can provide time-series dataset on PM$_{2.5}$ components, the additional organic aerosol mass spectra information obtained from the ToF-ACSM (Fröhlich et al., 2015), along with its enhanced capabilities for source apportionment, distinguishes it significantly from traditional techniques.

The organics dominated NR-PM$_{2.5}$ by 59%, comparable to that observed in nearby Lanzhou (56%), however generally higher than the 36-57% reported in NCP (shown in Fig. 5). This suggests a consistently greater contribution of organic aerosols in northwestern cities. The elevated fraction of organics may be due to the relatively low RH in arid and semi-arid regions, which limits the formation of SIA, as well as stronger and more prolonged solar radiation that facilitates organic aerosol formation. Together, these factors explained the high proportion of organics in northwestern cities. The latter explanation is further supported by the predominance of SOA in Yinchuan, as discussed in Sec. 2.3. Sulfate constituted 16% of NR-PM$_{2.5}$,



making it the largest component within SIA, followed by nitrate at 13%. The relatively low proportion of summertime nitrate is anticipated due to the strong evaporation of ammonium nitrate under high temperatures. However, we found that when NR-PM$_{2.5}$ exceeded 25 µg m$^{-3}$, nitrate surpassed sulfate to become the dominant SIA species, and its contribution increased with rising pollution levels. This trend highlights the growing significance of nitrate in particulate pollution, even though in summer

Yinchuan, consistent with observations conducted in Beijing during fall and winter (Zhou et al., 2019).

The entire campaign was divided into two phases according to meteorological conditions. As shown in Fig. 6, Period 1 (P1, 6-21 June) was characterized by relatively low RH (31±14%) and strong solar radiation, while Period 2 (P2, 22 June -10 July) was marked by higher RH (56±17%) due to frequent rainfall. Concurrently, the concentration of NOx decreased by 46%, from 29.1 µg m$^{-3}$ in P1 to 15.7 µg m$^{-3}$ in P2, indicating substantial changes in primary emissions between the two phases. In

response, the mass concentrations of organics exhibited the most pronounced decline, decreasing by 48.5% from P1 to P2, with mass fractions dropping from 69% to 48%. A similar reduction of 26.5% was observed for chloride, following the trend of organics as a response to decreased primary emissions. In contrast, the concentrations of SIA species showed unexpectedly 13.3-27.2% increases during P2 compared to P1, with nitrate showing the most significant rise. Overall, the total NR-PM$_{2.5}$ mass decreased by 5.3 µg m$^{-3}$ from P1 to P2, with the contribution of SIA increasing from 30% to 41%, accompanied by

decreases in organics and chloride. These results underscore the considerable influence of meteorology on rapid changes in aerosol compositions, highlighting contrasting effects on primary and secondary components associated with their distinct sources and formation mechanisms. While high RH during P2 promotes the formation of secondary species, it also enhances wet scavenging. The effects of aqueous-phase processing can offset the losses associated with wet removal, ultimately contributing to an increase in secondary species. Although previous studies have revealed the dominant role of light rain in the

accumulated wet removal of aerosols (Wang et al., 2021), the frequent rainfall in this study facilitated the production of secondary species. However, the decreases in primary species can be attributed to the combined effects of wet removal and a decline in primary emissions. These results differ significantly from those reported by Li et al. (2023b), which indicated an increased contribution of semi-volatile species such as nitrate and chloride, due to enhanced gas-particle partitioning, along with a decreased contribution of sulfate and organics based on long-term statistical analysis in summer Beijing.

In addition to the total mass concentrations, aerosol species correspondingly exhibited significant differences in their diurnal variation patterns between the two phases (Fig. S1). For instance, sulfate began to rise in the early morning and peaked around 12:00 during P1; whereas during P2, it continued to increase after noon, reaching a maximum at 17:00. This change might be due to enhanced aqueous-phase processing of sulfate during P2. It is note that traffic-related NOx emissions led to a peak in nitrate between 09:00 and 10:00, highlighting the considerable role of traffic emissions in Yinchuan. Furthermore,

nitrate concentrations were even 50-100% higher than in P1, particularly at 00:00-05:00, despite the lower NOx levels during P2. This increase in nocturnal nitrate was possibly due to the heterogeneous hydrolysis of N$_2$O$_5$ under high RH conditions (Yan et al., 2023), which could also account for the overall higher nitrate levels observed in P2.

**3.3  Sources of organic aerosols**





The PMF analysis was applied to the OA matrix measured by the ToF-ACSM, resulting in the identification of one POA and two SOA factors through spectral profile characterization and analysis with precursor species. The POA was characterized by typical hydrocarbon ions, including the $C_nH_{2n+1}^+$ and $C_nH_{2n-1}^+$ categories (e.g., $m/z$ 27, 41, 43, 55, 57), which primarily originate from primary emission sources such as coal combustion, vehicle exhaust, and cooking activities (Zhang et al., 2007). Distinct peaks at $m/z$ 91 and $m/z$ 115, indicative of polycyclic aromatic hydrocarbons (PAHs)(Xu et al., 2020), were observed in this study, suggesting the significant influence of coal combustion in Yinchuan, even during the summertime. This finding contrasts with conditions in summer NCP, where coal combustion is typically considered to be negligible in non-heating seasons due to a strict ban on coal burning. Although the POA exhibited a high $m/z$ 55/57 ratio, which is often regarded as an indicator of cooking emissions(He et al., 2010;Zhang et al., 2021), there were no pronounced cooking-related peaks during meal times (e.g., 12:00 and 20:00). A possible explanation for this discrepancy is that the sparse population in Yinchuan results in lower emissions from the dining sector. Instead, two prominent POA peaks occurred at 08:00-09:00 and 22:00-23:00. The small morning peak corresponded to traffic rush hour, consistent with the diurnal variation of NOx, which also peaked in the morning. This interpretation is further supported by the concurrent morning peaks of benzene and toluene, which commonly serve as tracers for vehicle emissions. We observed pronounced contrast in the diurnal variations of POA between P1 and P2. During P1, POA displayed significant morning peaks, which was considerably weaker or absent in P2. This difference may be attributed to lower primary emissions during P2, as indicated by the concurrently decreased NOx levels. On the contrary, a much higher late-evening peak likely corresponded to industrial emissions from nighttime operations. This elevation in concentration was further amplified by the shallower nocturnal boundary layer during the night. Despite this, it should be noted that, due to the limited mass resolution of ToF-ACSM, specific sources could not be unambiguously distinguished. The resolved POA factor often represented a mixture from multiple sources. In this study, the mixed POA was interpreted as encompassing emissions from traffic, coal combustion and industrial activities. Overall, POA comprised an average of 26% of the total OA, with a higher contribution in P1 than P2 (29% vs. 21%) in response to enhanced primary emissions during P1. Note that the contribution of POA could increase to 40% at nighttime, particularly during P1, indicating the significant influence of industrial activities on organic aerosol compositions in Yinchuan.

The two secondary factors were marked by prominent peaks at $m/z$ 28 and $m/z$ 44, which are representative of $CO_2^+$. The oxygenated organic aerosol (OOA) showed a distinct daytime increase after 07:00, indicative of photochemical production (Xu et al., 2017). The afternoon decline of OOA concentrations was primarily attributed to dilution effects associated with the rising planetary boundary layer. This reasoning is supported by the continuous increases in the mass fractions of OOA from early morning to late afternoon. In contrast, the other SOA factor displayed a relatively flat concentration profiles throughout the day, suggesting a regional background source. Thus, it was classified as regional-related SOA, corresponding to its mass fractions remaining nearly constant at 10%. We found significantly different mass spectral and diurnal patterns between two SOA factors, resulting from their distinct sources and formation mechanisms. Compared to regional-related SOA, OOA exhibited a higher oxidation degree, as reflected by its higher $f_{44}$. Instead, the mass spectral profile of regional-related SOA was characterized by a high $f_{29}$, indicating that it may come from similar source as POA. For example, the regional-related



SOA also showed a small morning peak similar to that of POA. Consistently, a strong correlation was observed between regional-related SOA and POA, with a R$^2$ of 0.67. These findings suggest that the regional-related SOA in this study likely
originates from regional-scale oxidized primary emissions. It is important to note that when comparing the proportion of $f_{44}$, SOA in Yinchuan is highly more oxidized than those at other sites in the NCP, possibly due to prolonged sunlight exposure during the summertime.

On average, SOA (=OOA + regional-related SOA) accounted for 74% of total organic aerosols, underscoring the predominance of SOA in summer Yinchuan. This finding aligns with Li et al. (2023a) reporting that SOA represented 85-90%
of OA in Beijing during the five summers from 2018 to 2022. This dominance is not surprising, given that POA primarily originates from vehicle emissions and cooking, with minimal contributions from coal combustion and biomass burning during the summer months. In comparison to Beijing, Yinchuan exhibited nearly double the contribution of POA, ascribing to the significant impact of industrial activities in the region. These results indicate a greater contribution of primary-related emissions in northwestern China than Beijing. While meteorological conditions had a substantial influence on SIA species,
leading to increased mass concentrations of SIA species under elevated RH conditions, all organic aerosol components showed consistent decreases from P1 to P2. Among the OA categories, POA experienced the largest decline of 61%, and the two SOA factors displayed comparable decreases of 41-44%. As a result, the mass fraction of SOA increased to some extent from 71% at P1 to 78% in P2. Note that the more substantial reduction in POA than SOA can be resulted from additional decreases in primary emissions during P2, alongside the influence of meteorological factors. Interestingly, the composition of SOA
remained relatively stable between P1 and P2, with OOA consistently dominating at 86-87%, indicating a similar meteorological influence on different SOA sources in this study. Our results demonstrate the differing behaviors of inorganic and organic aerosols in response to meteorological conditions, potentially due to their distinct formation mechanisms. For instance, SIA was primarily generated through aqueous-phase processing in this study, and thus elevated RH during the transition from P1 to P2 promoted their formation. In contrast, SOA was mainly produced from photochemical processes, and
reduced sunlight during P2 hindered their formation.

### 3.4  Variations of predominant VOCs precursors

We identified a total of 1118 chemical formulas for VOCs in this study. The CH category, which includes traditional GC-MS measurements of alkanes, alkenes, aromatics and terpenes, represented 24% of the total VOCs mass concentrations (Fig. S2). Instead, the CHO category comprised 74% of the VOCs, with a predominant contribution from species containing 1-2 oxygen
atoms, attributed to the Vocus PTR-MS's capability to detect low oxygen-containing species. Notably, as the contribution of SOA increased from P1 to P2, the proportion of CHO species with 3-5 oxygen atoms also rose from 10% to 16%. In fact, despite a sharp decrease in the mass concentrations of the CH category from 11.2 ppb to 8.3 ppb between P1 and P2, along with reductions in organic aerosols responsible from primary emissions, the total amount of CHO only declined by 0.5 ppb, with CHO species containing 3-5 oxygen atoms actually increasing. This trend might help explain the rising levels of SOA
during P2 under conditions of elevated RH, ascribing to favorable conditions for the formation of highly oxidized oxygenated





compounds.

We further explored some VOCs precursors to examine the potential link between variations in OA compositions and VOCs. Figure 9a suggests a strong correlation between the CH category and POA, featuring commonly recognized compounds such as benzene, toluene, xylene from vehicle exhaust. Additionally, propene, butene, pentene also exhibited highly
correlations with POA ($r$=0.67-0.72), as they are similarly emitted from buses. Correspondingly, these species showed prominent peaks during morning traffic hours. Different from isoprene, which reaches maximum concentrations during the daytime, we observed that monoterpenes and sesquiterpenes displayed higher concentration at night, peaking between 06:00 and 07:00. This suggests that terpenes have significant anthropogenic sources in urban areas, e.g., industry and traffic, in addition to direct emissions from vegetations in forests (Li et al., 2020). Not surprising, the diurnal patterns of terpenes closely
resembled those of methyl mercaptan, an important sulfur-containing component from vehicle exhaust. Note that typical oxidation products of monoterpenes such as $C_9H_{14}O$ and $C_{10}H_{16}O$, resulting from ozonolysis and reactions with OH·, also showed significant morning peaks. This highlights the considerable contribution of photochemical productions even during early morning hours when oxidant levels are low. We also found a range of nitrogen-containing species, including amines and urea, that were highly correlated with POA. Specifically, the contributions of the total CHN and CHON categories increased
to 5% in the morning, compared to only 3% during the day. These results indicate that, aside from NOx, traffic emissions contribute a non-negligible amount of organic nitrogen species, which are strongly associated with the formation of primary organic aerosol, particularly during morning traffic hours.

In addition, vanillin ($C_8H_8O_3$), a type of guaiacol derivatives typically linked to emissions from combustion process (Zhou et al., 2024), showed significant diurnal variations, with higher concentrations observed at night and in the early morning. This
finding further supports the combined effects of nocturnal burning sources and traffic emissions on POA. In comparison to POA, two SOA factors showed moderate correlations with a series of oxidation products of isoprene (Fig. 10), e.g., $C_4H_6O$, $C_5H_8O$, $C_5H_8O_4$ and $C_5H_8O_5$, with correlation coefficients ($r$) of 0.4-0.5. These species displayed pronounced increases during daytime, particularly after 08:00, peaking at early noon, which were responsible for the overall elevation of SOA throughout the day. This indicates the substantial role of isoprene-related products in daytime organic aerosols during summertime in
Yinchuan, attributable to significant biogenic isoprene emissions in summer. Beyond biogenic sources of SOA, we noticed substantial increases in the aromatic oxidation products during the daytime, such as benzaldehyde and hydroxyisophthalic acid ($C_8H_6O_5$), likely suggesting that traffic emissions undergo further oxidation to contribute to SOA formation. However, the relative contributions of traffic emissions to both POA and SOA should be further investigated.

### 3.5 Potential source regions of particulate pollution

The HYSPLIT model was utilized to calculate the 72-h back-trajectories of air masses, resulting in the identification of four distinct source region clusters. Throughout the observation period, 26% of the arriving air masses originated from the southeast (cluster C1). This cluster was traced back to central Shaanxi province, passing through northeastern Gansu province before reaching Yinchuan. It was characterized by warm, humid air masses, which significantly contributed to the presence of SIA



components, accounting for 42% of the total NR-PM$_{2.5}$ mass. The predominant source region identified during this campaign
was the northwest (C3), which represented 41% of the air masses observed. Interestingly, the three clusters (C1, C2, C3)
originating from the southeast, northeast and northwest source regions, displayed overall similar aerosol compositions, with
average NR-PM$_{2.5}$ concentrations ranging from 15.1 to 18.4 µg m$^{-3}$. Notably, the organics dominated NR-PM$_{2.5}$ by 57-60%,
followed by sulfate (15-17%) and nitrate (12-14%). These findings indicate a consistent nature of aerosol components on a
regional scale surrounding Yinchuan city, emphasizing the pronounced regional characteristics of air pollution within the
Yinchuan metropolitan area.

In contrast, Cluster C4, originating from Inner Mongolia and the northern part of Ningxia Hui Autonomous Region, was
characterized by long-range transport and exhibited a markedly different aerosol composition compared to the other clusters.
The proportion of SIA components sharply decreased to 18%, while the share of organic aerosols experienced a dramatic
increase to 80%. This shift can be attributed to the relatively dry air from the north, which hindered the formation of SIA.
However, organic aerosols underwent further oxidation and aging during the process of transport, consequently leading to their
increased mass concentrations. The different responses of SIA and organic aerosols to long-range transport demonstrate their
distinct formation mechanisms. For instance, SIA might undergo vaporization losses during transit, whereas organic aerosols
continued to oxidize persistently, which likely contributed to the observed relatively higher proportion of SOA in Yinchuan to
some degree.

In addition, it is noteworthy that the long-distance Cluster C4 passed through Shizuishan city, a typical heavy industrial
city in northern Yinchuan, indicating the non-negligible influence of localized primary emissions on air quality of urban
Yinchuan. As a result, the average NR-PM$_{2.5}$ concentrations during C4 reached a peak of 22.2 µg m$^{-3}$, slightly higher than the
levels observed during Cluster C1 to C3. This finding highlights the relatively significant role of long-distance transport in
contributing to particulate pollution in Yinchuan compared to nearby cities during the summer months, despite the common
perception that long-distance transport is more prevalent in winter. Consistently, during this campaign, the contribution of
Cluster C4 accounted for only 5% of the time. Upon further examination of the air masses, we found that cluster C4 primarily
occurred on 16-17 June, coinciding perfectly with wind directions originating from the north. In conclusion, our results reveal
that during the summertime, Yinchuan is predominantly influenced by homogeneous nearby air masses for the majority of the
time, with only a minor contribution from long-distance transport, even though the latter is associated with more polluted air
masses.

We also calculated the potential source contribution functions for aerosol compositions and OA factors (Fig. S3). Our
analysis revealed significant differences in the potential source regions for SIA and organic aerosols. The potential source
regions for SIA were primarily concentrated in a narrow vicinity around Yinchuan, specifically including Wuhai city, Alxa
League, and the Mu Us Sandy Land, all located approximately 300 km from Yinchuan. In comparison, organics were sourced
from greater distances, encompassing locations such as Lanzhou to the southwest and Ordos to the southeast. These differing
sources regions partially explain the varied responses of aerosol species to meteorological changes observed from P1 to P2.
For example, while SIA demonstrated a more pronounced sensitivity to local atmospheric conditions, organic aerosols





exhibited a greater influence from more distant sources. Overall, during this campaign, we identified three primary transport pathways for pollution affecting Yinchuan, that is, nearby cities surrounding Yinchuan as the main contributor, along with a southwestern pathway from Lanzhou and a southeastern pathway from Ordos and southern Ningxia. This understanding of transport pathways is crucial for developing effective air quality management strategies in the region. By identifying the spatial distribution of aerosol sources, we can better address the challenges of air pollution in Yinchuan and its surrounding areas.

## 4    Conclusions

This study provides new understanding of the evolving summertime $PM_{2.5}$-$O_3$ relationship and aerosol chemistry in Yinchuan, northwest China. Over the past decade, $PM_{2.5}$ levels declined substantially before 2019 and then stabilized, while $O_3$ rose and later plateaued, reflecting a transition from rapid emission-driven improvement to a regime dominated by photochemical processes. The weakening correlation between $PM_{2.5}$ and $O_3$ indicates that further particulate reduction alone is insufficient to mitigate ozone formation under the enhanced oxidative environment of cleaner air. Real-time ToF-ACSM and PTR-MS measurements showed that fine particles were mainly composed of organics, accounting for roughly 60% of the non-refractory $PM_{2.5}$ mass. SOA contributed nearly three-quarters of total organics, revealing that aged, photochemically processed material dominates Yinchuan's summer aerosol burden. In contrast, POA was largely linked to traffic and industrial activities, with additional input from sulfur- and nitrogen-containing compounds. VOC observations identified both anthropogenic and biogenic precursors, including aromatics, terpenes, and isoprene oxidation products, as key drivers of SOA formation. Air-mass analyses confirmed that local recirculation governs most summertime pollution events, with minor influence from long-range transport. Overall, our results highlight a regional shift toward SOA-dominated fine particles and decoupled $PM_{2.5}$–$O_3$ dynamics, underscoring the importance of coordinated control of VOCs, $NO_x$, and oxidants to achieve sustained air-quality improvement in northwestern China.

### Data availability

Data used in this study can be accessed from the Zenodo repository at https://doi.org/10.5281/zenodo.17784732 (Zhou 2025).

### Author contribution

WZ, JL and YS designed the research. WZ, LY, SZ conducted the measurements. WZ, LY, SZ, YK, LY, WZ, WW, ZZ, YL, WX and YS analyzed the data. YG, YW, ZZ, JL, ZW and YS reviewed and commented on the paper. WZ, JL and YS wrote the paper.

### Competing interests

The contact author has declared that none of the authors has any competing interests.

### Acknowledgements

This research was supported by grants from the Key Research and Development Program of Ningxia (2024BEG01002).





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



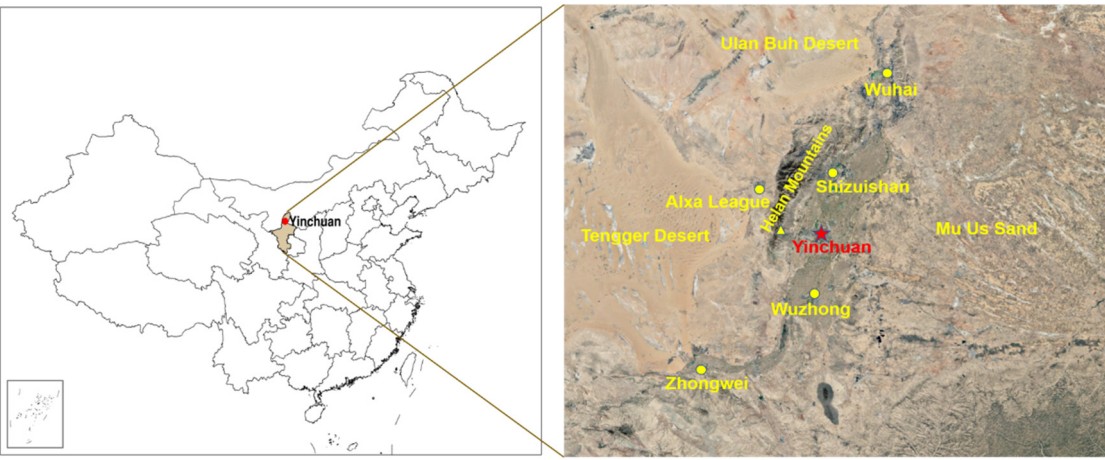

**Figure 1: Location of the sampling site and surrounding cities.**

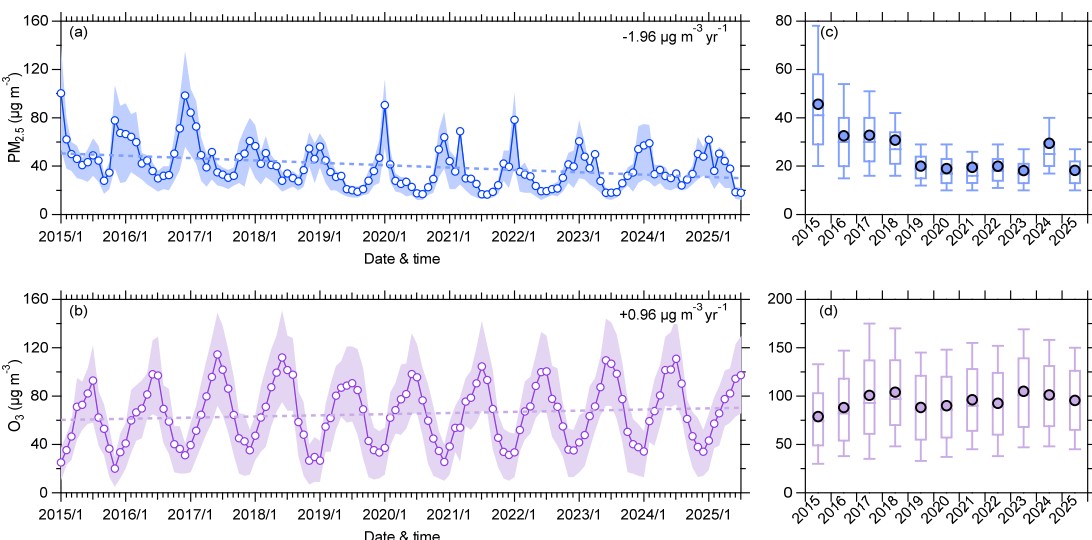

**Figure 2: (a-b) Monthly variations of PM$_{2.5}$ and O$_3$ in Yinchuan from 2015 to 2025, with shaded areas representing the 25$^{th}$ and 75$^{th}$ percentiles. Also shown are the linear interannual changes rate of PM$_{2.5}$ and O$_3$. (c-d) Box plots of average summertime PM$_{2.5}$ and O$_3$ concentrations. The circles, horizontal lines, lower and upper box edges, lower and upper whiskers refer to mean, median, 25$^{th}$ and 75$^{th}$ percentiles, 10$^{th}$ and 90$^{th}$ percentiles in the box, respectively.**





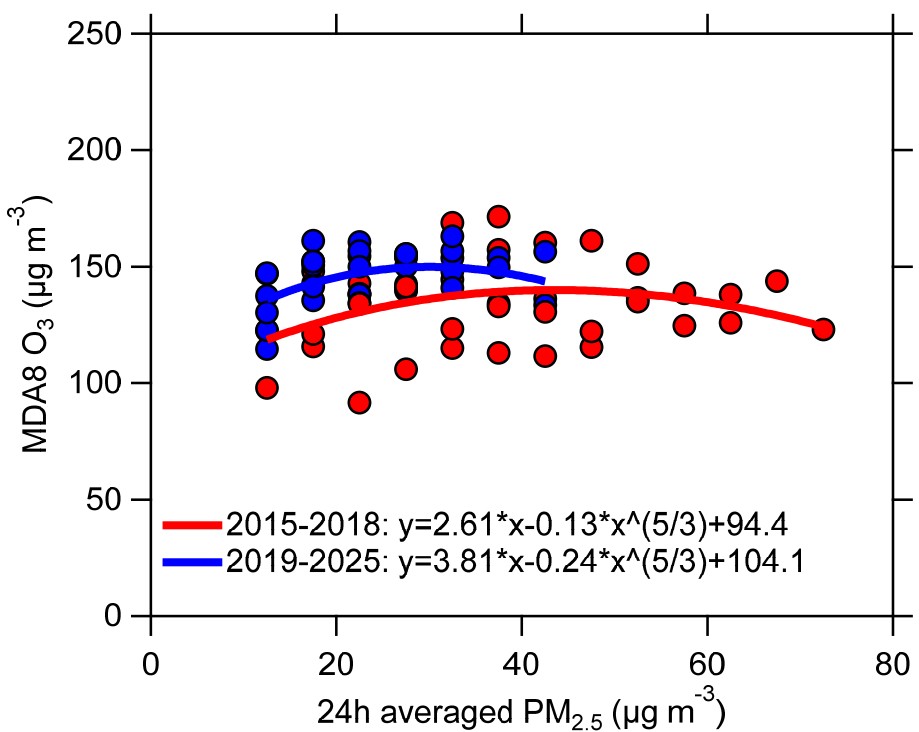

**Figure 3: Relationship between 24-h averaged PM₂.₅ and MDA8 O₃ fitted using an exponential equation for the summertime months during 2015-2025.**

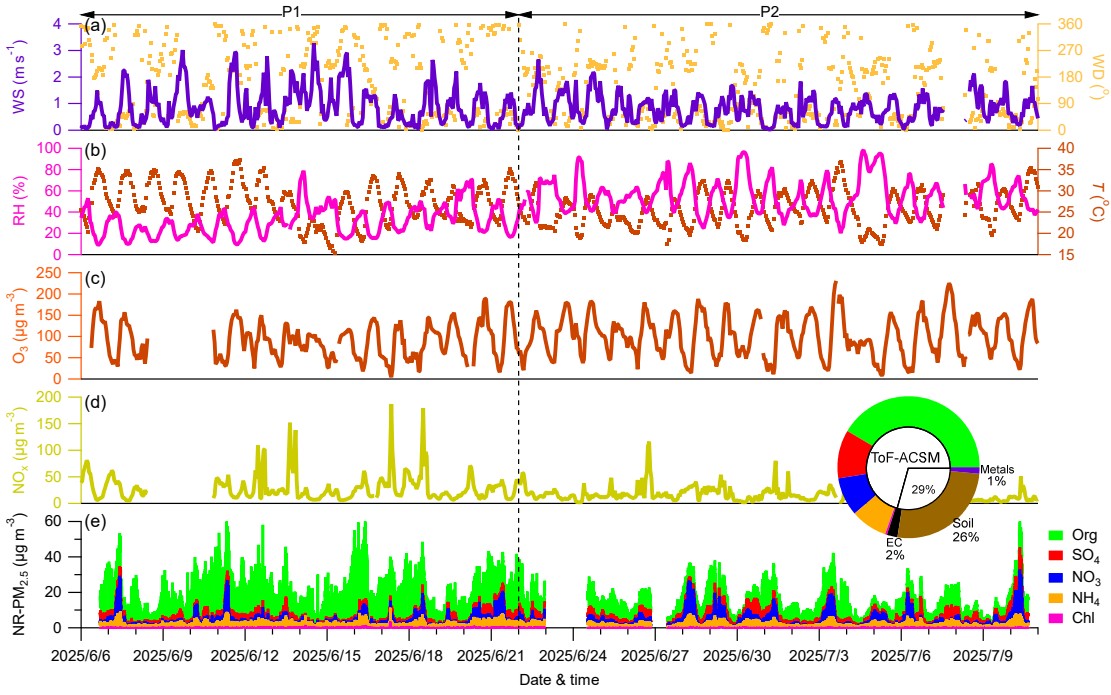





**Figure 4: Time series of (a-b) meteorological variables (WS, WD, RH and *T*), (c-d) gaseous species (O₃ and NOx), (e) chemical species for ToF-ACSM. The pie chart shows the average compositions of particles including organics, sulfate, nitrate, ammonium, chloride, elemental carbon, soil and metals for the entire campaign.**

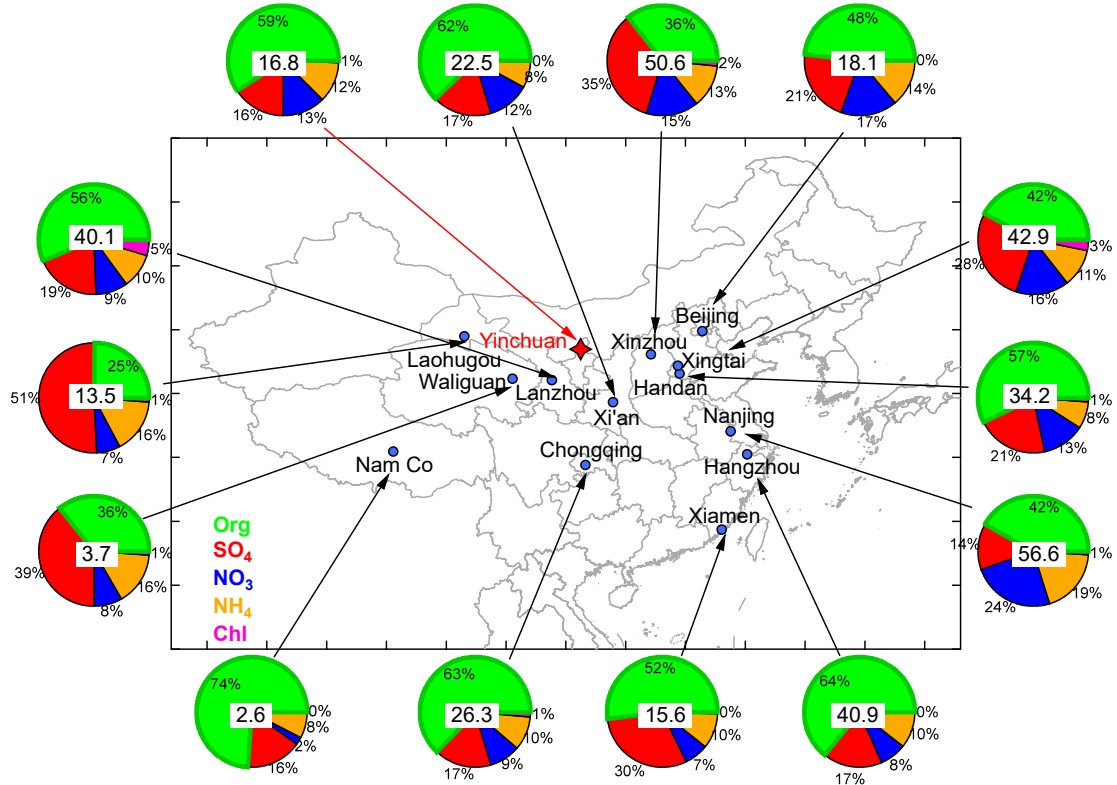

590

**Figure 5: Average mass concentrations and chemical compositions of total NR-PM₂.₅ mass loadings observed in China using different types of aerosol mass spectrometers (e.g., ToF-ACSM, Q-ACSM, HR-ToF-AMS) during the summertime.**



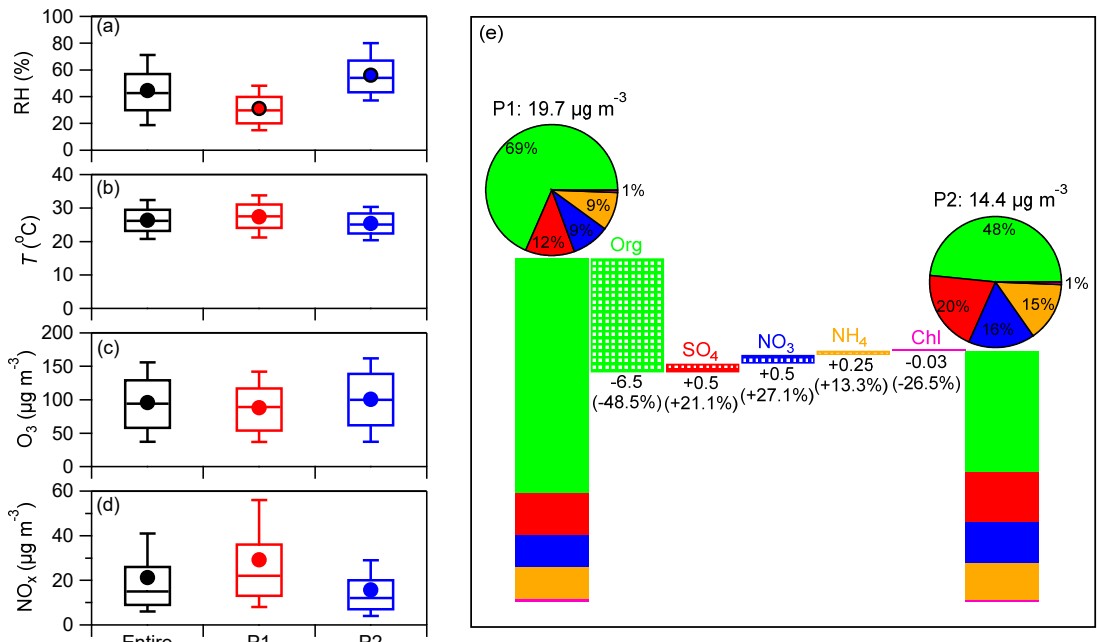

**Figure 6: (a-d) Box plots of average meteorological variables (RH, _T_) and gaseous precursors (O₃, NOx) during the entire campaign, P1 and P2. The circles, horizontal lines, lower and upper box edges, lower and upper whiskers refer to mean, median, 25th and 75th percentiles, 10th and 90th percentiles in the box, respectively. (e) Average mass concentrations and changes of NR-PM₂.₅ components from P1 to P2, along with average compositions during P1 and P2 shown in pie charts.**

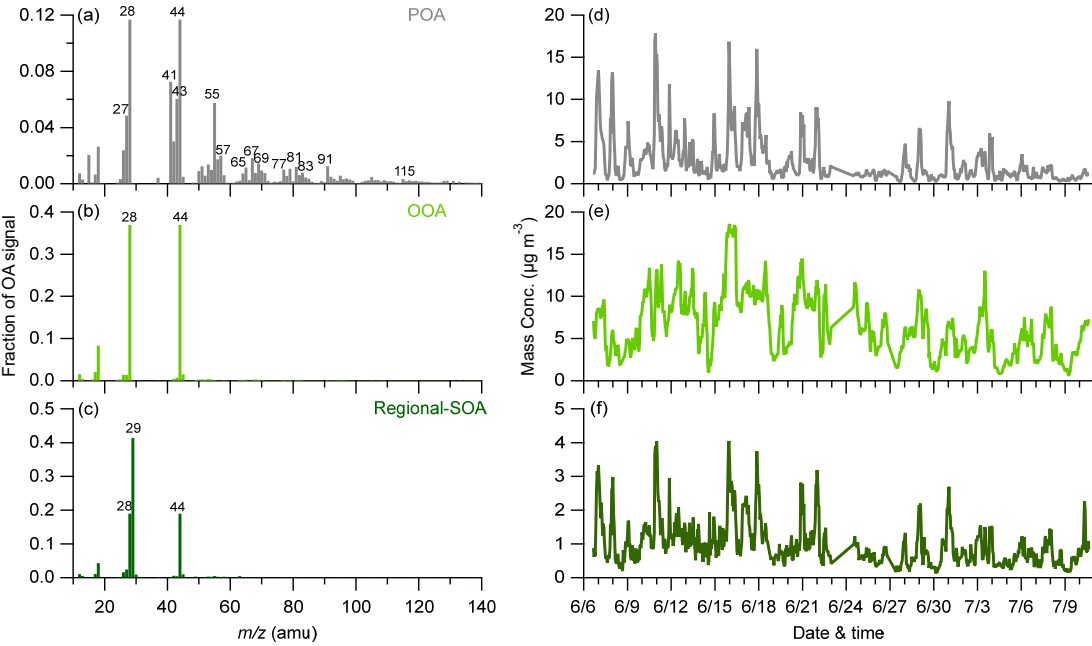

**Figure 7: (a-c) Average mass spectral profiles and (d-f) time series of OA factors (POA, OOA, Regional-SOA) during the entire**



**campaign.**

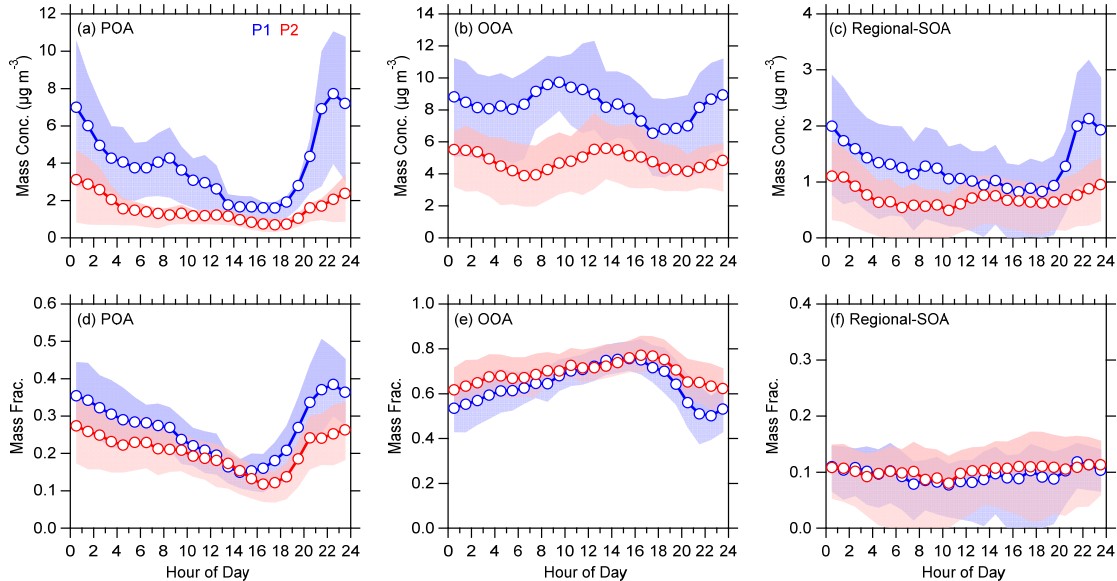

**Figure 8: Diurnal variations of OA factors (POA, OOA and Regional-SOA) during P1 and P2, respectively. The circles indicate mean values, and the shaded areas represent the 75th and 25th percentiles.**





**Figure 9: (a) Correlations between POA and VOCs, and diurnal patterns of partial POA-related VOCs, including (b) $C_6H_6$, (c) $C_8H_8O_3$, (d) $C_3H_5N$, (e) $CH_4S$, (f) $C_{10}H_{16}$, (g) $C_{15}H_{24}$. The circles indicate mean values, and the shaded areas represent the 75th and 25th percentiles.**





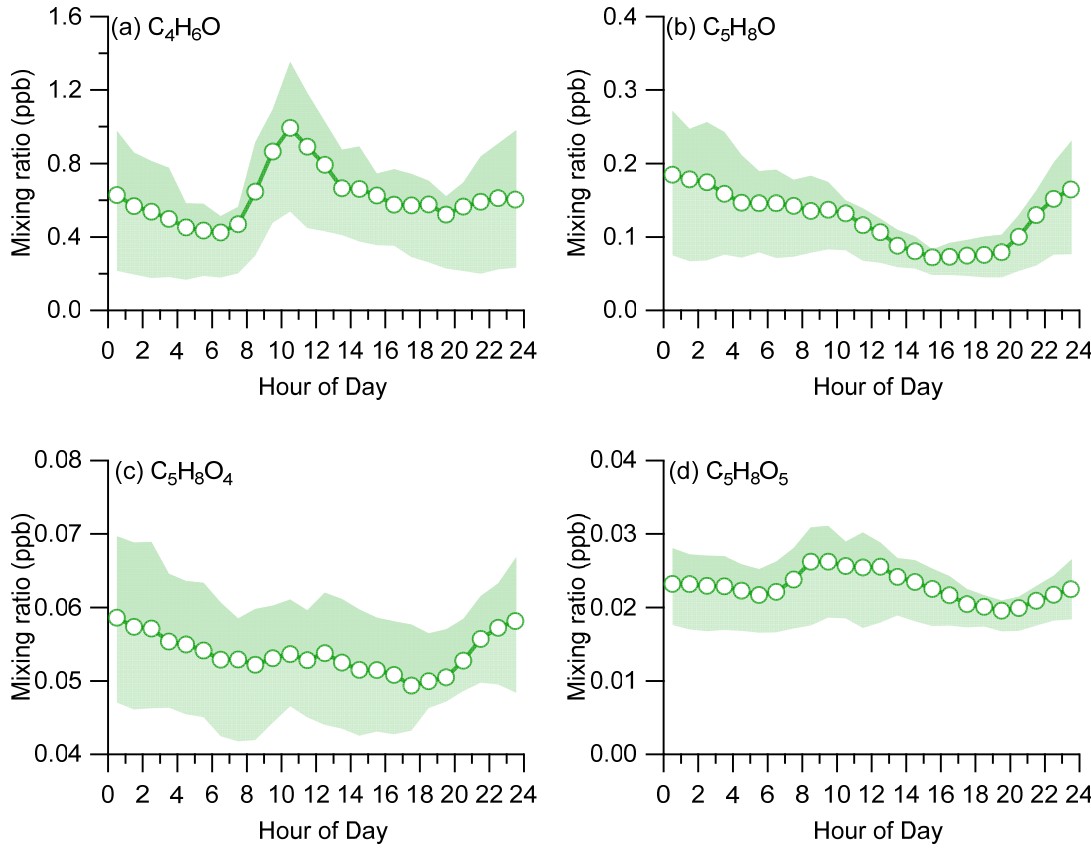

**Figure 10: Diurnal patterns of isoprene oxidation products, including (a) C$_4$H$_6$O, (b) C$_5$H$_8$O, (c) C$_5$H$_8$O$_4$, (d) C$_5$H$_8$O$_5$. The circles indicate mean values, and the shaded areas represent the 75$^{th}$ and 25$^{th}$ percentiles.**





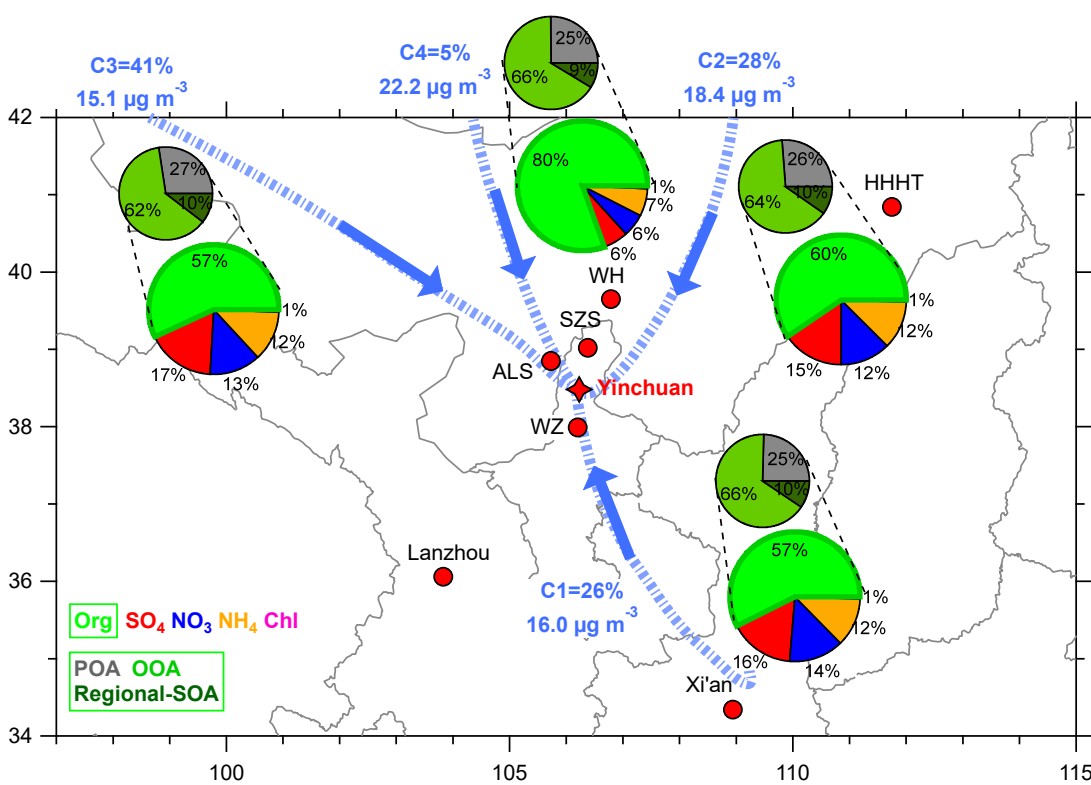

**Figure 11: Back trajectories of air mass arriving at Yinchuan during the campaign. The pie charts show the average NR-PM2.5 and OA composition for each cluster. Also shown are the percentages of the total trajectories, and corresponding average mass concentration of NR-PM2.5 for each cluster.**