# Peer review of "Decadal Transition of Summertime PM2.5-O3 Coupling and Secondary Organic Aerosol Dominance in Northwest China"

_EGUsphere, 2025_

## Author Comment (AC1)

We are thankful to the two reviewers for their insightful and constructive comments, which help us significantly improve the quality of our manuscript. We have carefully revised the manuscript accordingly. Below, we provide our point-to-point responses to each comment in blue, with the changes in revised text highlighted in underlined format.

**Response to Reviewer #1**

Zhou et al. investigated the long-term trends of particulate matter pollution in a typical city in Northwest China by analyzing a decade of observational data, focusing on the source apportionment of SOA. They highlighted the significant role of local aging processes of particulate matter in the synergistic pollution of $PM_{2.5}$ and $O_3$. Overall, this manuscript is well-written. I recommend its publication in ACP after addressing the following minor comments.

1.This study uses a single field campaign to demonstrate the importance of local particulate matter aging in the $PM_{2.5}$-$O_3$ pollution. Why then analyze $PM_{2.5}$ and $O_3$ data from the past ten years? Is the single campaign representative? Could this core viewpoint potentially be invalidated over a decadal timescale?

We thank the reviewer for these insightful questions. This study focuses on $PM_{2.5}$-$O_3$ co-pollution in Yinchuan. While the relationship between $PM_{2.5}$ and $O_3$, as well as their long-term trends, have been extensively documented in major metropolitan regions, e.g., the NCP, YRD and PRD. Such analyses remain scarce in northwestern China. To address this gap, our study first examines a decade-long dataset of $PM_{2.5}$ and $O_3$ to characterize the historical pollution patterns in Yinchuan.

Building upon this long-term assessment, we conducted a comprehensive field campaign to measure the chemical composition of $PM_{2.5}$ and $O_3$, along with their key precursors-VOCs, to further elucidate the sources and formation mechanisms of $PM_{2.5}$ in this region.

Our campaign was conducted at the Yinchuan Atmospheric Environmental Super Monitoring Station, which represents a typical urban environment in the city. The application of ToF-ACSM and Scout PTR-ToF-MS have been widely validated in previous studies for charactering aerosol composition and VOCs speciation. The representativeness of the sampling site, combined with the robustness of these measurement techniques, ensures that our findings are reflective of urban air quality in Yinchuan

Admittedly, a single intensive campaign cannot fully capture interannual variability driven by changes in emission or meteorology. However, the lack of comparable datasets from prior years limits direct comparison with the past decade. Despite this, given that major emission sources and industrial structures in Yinchuan, such as traffic and industrial activities, have remained relatively stable over the short term, our observations provide a representative snapshot of recent pollution characteristics and sources of organic aerosol. For example, our results indicate that SOA formation is

driven by a combination of anthropogenic and biogenic precursors, including urban terpenes and aromatic oxidation products from traffic and industry emissions.

2. Lines 56-70: The logic connecting this paragraph to the research content of this paper is not well established. It is suggested that the authors revise this paragraph from the following perspectives. For instance, what are the current differences in the sources and formation mechanisms of SOA across different regions of China? What is the necessity of using AMS (or ACSM) for this research? Why focus on the sources and formation mechanisms of SOA in Northwest China?

We sincerely appreciate the reviewer for these valuable suggestions. The sources and formation mechanisms of SOA across China are strongly influenced by precursor types and prevailing meteorological conditions. For instance, the relatively high humidity in southwest China facilitated the formation of SOA, which contrasts sharply with the much drier climate of northwest China.

Compared to well-studied regions such as the NCP, YRD and PRD, research on northwestern China is still insufficient. The unique geographical features and proximity to dust source regions may lead to significantly different characteristics of $PM_{2.5}$ pollution in northwest China compared to other areas in China. Thus, this study specifically focuses on elucidating the sources and formation processes of SOA in northwestern China.

In contrast to traditional offline methods and online ion chromatography, AMS or ACSM techniques can not only enable real-time online measurements of aerosol compositions with high time resolution (several minutes), but also provide additional organic aerosol mass spectra information. These capabilities greatly enhance the accuracy and robustness of source apportionment analyses.

We have rephrased these sentences in the revised manuscript to improve clarity and fluency,

"Previous field campaigns on atmospheric organic aerosols in China have primarily focused on three major metropolitan regions, e.g., the North China Plain (NCP), Yangtze River Delta (YRD), and Pearl River Delta (PRD)(Zhou et al., 2020). More recently, several measurements have expanded to the Fenwei Plain in the Yellow River Basin, including Yuncheng, Shanxi province (Li et al., 2022;Wang et al., 2024) and Baoji, Shaanxi province (Li et al., 2025), as well as the Chengdu-Chongqing Basin in southwest China (Bao et al., 2023). These studies reveal significant spatial heterogeneity in OA sources and compositions, driven by variations in precursor emissions, meteorology, topography and local industries. For instance, high humidity in southwest China promotes the formation of SOA, which contrasts sharply with the arid conditions in northwest China. Among current online analytical techniques, aerosol mass spectrometry (AMS) has become the method of choice in field campaign, owing to its ability to provide real-time, high time-resolution measurements of aerosol compositions and detailed organic mass spectra information. These capabilities greatly

enhance the accuracy and robustness of source apportionment. Despite growing observational efforts, research in northwestern China remains scarce. Xu et al. (2014) conducted the first AMS-based measurements in Lanzhou, a major city in northwest China, finding that local traffic and coal combustion dominated summertime OA, while non-fossil sources (e.g., biomass burning and cooking activities) contributed ~55% in winter (Xu et al., 2016). However, Lanzhou is situated in a narrow valley between two mountain ranges, and its unique topography limits its representativeness for the broader northwestern region, leaving OA characteristics across most of this vast area poorly understood."

3. Lines 120-122: More details regarding the PMF analysis based on ACSM spectra are needed to be provided. For example, how were the three source factors determined, and what are their corresponding source profiles?

Thanks for this good suggestion. The PMF results of ACSM spectra were selected based on a comprehensive evaluation, including the diagnostic plots from the PMF analysis, as well as the characterization and interpretation of OA source spectral profiles in conjunction with analysis with precursor species.

The spectral profiles have been given in detail in Sec. 3.3. POA was characterized by typical hydrocarbon ions, including the $C_nH_{2n+1}^+$ and $C_nH_{2n-1}^+$ categories (e.g., $m/z$ 27, 41, 43, 55, 57), which primarily originate from primary emission sources such as coal combustion, vehicle exhaust, and cooking activities. Distinct peaks at $m/z$ 91 and $m/z$ 115, indicative of polycyclic aromatic hydrocarbons (PAHs), were also observed in this study.

The two SOA factors were marked by prominent peaks at $m/z$ 28 and $m/z$ 44, which are representative of $CO_2^+$. Compared to regional-related SOA, OOA exhibited a higher oxidation degree, as reflected by its higher $f_{44}$. Instead, the mass spectral profile of regional-related SOA was characterized by a high $f_{29}$, indicating that it may come from similar source as POA.

We added the diagnostic plots of PMF analysis in the supplementary information,

[Figure]

Figure R1. The diagnostic plots of PMF analysis of ToF-ACSM.

4. Line 175: What does "second-phase pollution control measures" mean? Is it the same as "Phase II" mentioned later?

We thank the reviewer for raising this point. The "second-phase" pollution control measures refer to China's Three-Year Action Plan to Win the Battle for Blue Skies during 2018-2020, which was in reference to the first-phase control measures "Ten Measures for Air Pollution Prevention and Control" during 2013-2017.

In this study, "Phase I" and "Phase II" denote two distinct periods in Yinchuan that reflecting differing responses of $O_3$ and $PM_{2.5}$ to these national air pollution control strategies. Specifically, Phase I (2015-2018) was characterized by sharp reductions in $PM_{2.5}$ levels, coincided with significant increases in MDA8 $O_3$; however, Phase II (2019-2025) marks a period during which $O_3$ became less sensitive to further controls on $PM_{2.5}$, suggesting a shift in the coupling relationships between $PM_{2.5}$ and $O_3$.

5. Lines 182-185: What is the logical relationship between the viewpoint expressed in this sentence and the described $PM_{2.5}$-$O_3$ relationship in Phase I and Phase II mentioned earlier?

We thank the reviewer. Between 2013 and 2020, both $O_3$ and $PM_{2.5}$ experienced a two-stage evolutionary patterns-shifting from an initial phase of rapid increase or decrease to a subsequent phase characterized by more flattened trends. Recent studies by Wang at el. (2024) and Geng et al. (2025) have attributed these nonlinear responses of air pollutants to emission control measures to changes in atmospheric oxidation capacity and the influence of natural factors. These findings were cited to support the similar two-phase behavior in Yinchuan within the broader national trend, thereby reinforcing the plausibility and robustness of the distinct of $PM_{2.5}$-$O_3$ relationship identified in Phase I (2015-2018) and Phase II (2019-2025).

To enhance clarify and avoid potential confusion, we have expanded this discussion in the revised manuscript,

"A very recent study has also revealed similar trend of $O_3$ levels in Beijing as a response to national-wide emission controls during 2005-2020, characterized by an initial rapid increase followed by a gradual decrease, attributed to changes in the atmospheric oxidation capacity (Wang et al., 2024b). This is consistent with Geng et al. (2024) showing that the "Ten Measures for Air Pollution Prevention and Control" achieved significant emission reductions in $PM_{2.5}$ in China during 2013-2017, however, the potential for further reduction after 2017 has become limited based on emission inventories and model simulations. These studies supported the plausibility and robustness of the observed two-phase $PM_{2.5}$-$O_3$ relationship in Yinchuan within the context of national trend. Therefore, the timing of controlling VOCs and NOx is crucial keys for the coordinated reductions of both $O_3$ and $PM_{2.5}$ levels in Yinchuan, and even the synergistic tripe controls of $O_3$, $PM_{2.5}$ and $CO_2$ in the future (Liu et al., 2025)."

6. It is suggested to move Fig. 3 to the Supplementary Information.

The long-term annual average concentrations of $PM_{2.5}$ and $O_3$ shown in Fig. 2 reveal that the relationships between these two pollutants evolved through two distinct stages, Phase I (2015-2018) and Phase II (2019-2025). On this basis, the fitting curves in Fig. 3 directly illustrate that $O_3$ has become less sensitive to changes in $PM_{2.5}$ in Yinchuan. Also, the earlier timing of the inflection points in Phase II compared to Phase I, suggesting that continued efforts to reduce $PM_{2.5}$ are expected to more effectively manage $O_3$ levels over time.

Thus, Fig. 3 directly highlights the complex and evolving coupling between $PM_{2.5}$ and $O_3$. Retaining it in the main text better aligns with the core focus of this study: elucidating the long-term $PM_{2.5}$-$O_3$ coupling relationships in northwestern China. We still thank the reviewer's suggestions.

7. What is the purpose of Fig. 5? It seems the text does not discuss Fig. 5 in detail. Consider moving Fig. 5 to the Supplementary Information.

We thank the reviewer for this helpful suggestion. Figure 5 presents the average NR-$PM_{2.5}$ composition across major Chinese cities during summer, based on AMS or ACSM measurements. Figure 5 reveals notable differences in aerosol composition

among different regions. For example, the northwestern cities such as Yinchuan and Lanzhou showed consistently higher contributions of organic aerosols than the NCP, likely due to stronger solar radiation promoting the formation of secondary organic aerosols, as well as low RH suppressing the production of SIA species. Additionally, the average NR-PM$_{2.5}$ concentration in Yinchuan is significantly lower than those in the NCP and YRD.

We agree with the reviewer's recommendation to move Fig. 5 to the supplementary Information to make the main figures concise.

8. Lines 238-254: This paragraph is somewhat redundant. Its removal would not affect the overall logic of the paper. The authors may consider removing it.

In this study, we focus on the aerosol composition and sources of PM$_{2.5}$. However, the ToF-ACSM could only detect the non-refractory PM$_{2.5}$. Given that Yinchuan is frequently affected by dust events and suspended particulate matter, refractory components such as black carbon, soils and metals may also play a significant role. To completement the ToF-ACSM measurements, we estimated the average aerosol composition of total PM$_{2.5}$ in Yinchuan, which confirms that NR-PM$_{2.5}$ constitutes the majority of PM$_{2.5}$ during summertime.

We agree with the reviewer that the original paragraph is somewhat redundant. Accordingly, we removed the detailed comparisons among the ToF-ACSM, online ion chromatograph, and Sunset OC/EC analyzer to streamline the discussion, retaining only the analysis relevant to total PM$_{2.5}$ mass.

"On average, the NR-PM$_{2.5}$ concentrations measured by the ToF-ACSM accounted for 71% of the total PM$_{2.5}$ mass (=NR-PM$_{2.5}$+EC+Soil+Metals), indicating that the ToF-ACSM effectively captured the majority of fine particulate matter components in Yinchuan. Among the refractory species, soil constituted 26% of PM$_{2.5}$, while metals contributed 1%, primarily due to calcium originating from suspended dust and construction activities. In comparison to dust-free cities, the proportion of NR-PM$_{2.5}$ within total PM$_{2.5}$ was lower in this study, e.g., Fenhe Plain (Li et al., 2022) and central China Plain (Li et al., 2021b). However, this finding is consistent with the regional context, as northwestern China is prone to dust events. Although summertime is generally less susceptible to dust storms, nearby sand sources still contribute to elevated levels of suspended dust in Yinchuan, resulting in much higher presence of refractory species in this study than previous studies."

9. Lines 275-277: Please provide further explanation. How do meteorological conditions lead to rapid changes in aerosol chemical composition? Is there any relevant mechanism or literature support?

We thank the reviewer for this suggestion. The entire campaign was divided into two distinct phases based on prevailing meteorological conditions. Period 1 (P1, 6-21 June) was characterized by relatively low RH (31±14%) and strong solar radiation, while

Period 2 (P2, 22 June -10 July) was marked by higher RH (56±17%) due to frequent rainfall events.

These contrasting meteorological conditions led to pronounced differences in aerosol chemical composition between P1 and P2. Specifically, the elevated RH during P2 promoted significant increases in secondary inorganic aerosols by 13.3-27.2%, e.g., sulfate, nitrate and ammonium. The main formation pathway of SIA under RH is via aqueous-phase processing or heterogeneous reactions (Sun et al., 2013;Sun et al., 2014).

We elaborated this in the revised manuscript and added relevant references,

"These results underscore the considerable influence of meteorology on rapid changes in aerosol compositions, highlighting contrasting effects on primary and secondary components associated with their distinct sources and formation mechanisms, e.g., aqueous-phase processing, heterogeneous reactions or enhanced gas-particle partitioning (Sun et al., 2014)."

10. The source apportionment results based on ACSM data suggest that coal combustion still contributes to OA in summer. Given the availability of concurrent XRF data, could the authors provide further insights from the perspective of PM$_{2.5}$ source apportionment?

We thank the reviewer for this suggestion. XRF-derived crustal elementals can indeed provide information on primary PM$_{2.5}$ sources using CMB method, including the coal combustion, traffic and industrial emissions. However, CMB results are subject to uncertainties arising from the selection of source profiles, as primary emission spectra can vary significantly across different regions of China. In addition, the dataset of 27 crustal elementals contained considerable missing values, which further limits the reliability and applicability of source apportionment.

Moreover, While CMB is well-suited for apportioning crustal and primary elementals, it does not resolve SOA sources. In contrast, PMF applied to ACSM data enable detailed characterization of SOA processes, which dominated OA by 74% during summertime in Yinchuan.

Given that the primary objective of this study is to investigate the sources and formation mechanisms of OA, not general PM$_{2.5}$ source apportionment, and considering the uncertainties associated with XRF source apportionment, we opted to focus on the ACSM-based PMF approach. In future work, we will conduct XRF-based source apportionment to characterize PM$_{2.5}$ sources with a complete and robust XRF dataset.

11. Line 365: The full term for "r" (correlation coefficient) should be introduced here, not at Line 382. Furthermore, what is the difference between "r" and "$R^2$" at Line 247? This needs to be unified throughout the manuscript.

Thank you for pointing this out. We have corrected the notation a throughout the manuscript, consistently using the correlation coefficient $r$ instead of $R^2$. The revised sentences now read as below,

"Consistently, a strong correlation was observed between regional-related SOA and POA, with a correlation coefficient ($r$) of 0.82."

"…Additionally, propene, butene, pentene also exhibited strong correlations with POA ($r = 0.67$-$0.72$), as they are similarly emitted from buses."

"In comparison to POA, two SOA factors showed moderate correlations with a series of oxidation products of isoprene (Fig. 10), e.g., $C_4H_6O$, $C_5H_8O$, $C_5H_8O_4$ and $C_5H_8O_5$, with $r$ of 0.4-0.5."

**Reference**

Bao, Z., Zhang, X., Li, Q., Zhou, J., Shi, G., Zhou, L., et al.: Measurement report: Intensive biomass burning emissions and rapid nitrate formation drive severe haze formation in the Sichuan Basin, China – insights from aerosol mass spectrometry, Atmos. Chem. Phys., 23, 1147-1167, 10.5194/acp-23-1147-2023, 2023.

Li, H., Zhang, T., Su, H., Liu, S. X., Shi, Y. Q., Wang, L. Y., et al.: Factors affecting the different growth rates of $PM_{2.5}$: Evidence from composition variation, formation mechanisms, and importance analysis of water-soluble inorganic ions with case study in northern China, Atmos. Environ., 340, 120913, 10.1016/j.atmosenv.2024.120913, 2025.

Li, Y., Du, A., Li, Z., Li, J., Chen, C., Sun, J., et al.: Investigation of sources and formation mechanisms of fine particles and organic aerosols in cold season in Fenhe Plain, China, Atmos. Res., 268, 106018, 10.1016/j.atmosres.2022.106018, 2022.

Sun, Y., Wang, Z., Fu, P., Jiang, Q., Yang, T., Li, J., et al.: The impact of relative humidity on aerosol composition and evolution processes during wintertime in Beijing, China, Atmos. Environ., 77, 927-934, 10.1016/j.atmosenv.2013.06.019, 2013.

Sun, Y., Jiang, Q., Wang, Z., Fu, P., Li, J., Yang, T., et al.: Investigation of the sources and evolution processes of severe haze pollution in Beijing in January 2013, J. Geophys. Res.- Atmos., 119, 4380-4398, 10.1002/2014jd021641, 2014.

Wang, W., Cui, Y., Zhang, R., He, Q., Gao, J., Fan, J., et al.: Characteristics of the chemical processes of organic aerosols by time-of-flight aerosol chemical speciation monitor (TOF-ACSM) in winter in a site of Fenhe Valley, northern China, Atmos. Pollut. Res., 15, 102132, 10.1016/j.apr.2024.102132, 2024.

Xu, J., Zhang, Q., Chen, M., Ge, X., Ren, J., and Qin, D.: Chemical composition, sources, and processes of urban aerosols during summertime in northwest China: insights from high-resolution aerosol mass spectrometry, Atmos. Chem. Phys., 14, 12593-12611, 10.5194/acp-14-12593-2014, 2014.

Xu, J., Shi, J., Zhang, Q., Ge, X., Canonaco, F., Prévôt, A. S. H., et al.: Wintertime organic and inorganic aerosols in Lanzhou, China: sources, processes, and comparison with the results during summer, Atmos. Chem. Phys., 16, 14937-14957, 10.5194/acp-16-14937-2016, 2016.

Zhou, W., Xu, W., Kim, H., Zhang, Q., Fu, P., Worsnop, D. R., et al.: A review of aerosol chemistry in Asia: insights from aerosol mass spectrometer measurements, Environ. Sci. - Proc. Imp., 22, 1616-1653, 10.1039/D0EM00212G, 2020.

**Response to Reviewer #2**

The study of Zhou et al. investigates the changes in PM₂.₅ and ozone concentrations within a 10-year period in the Yinchuan metropolitan area of Northwest China. The authors combine complementary analytical approaches to characterize refractory and non-refractory fine-mode aerosol composition and relate these observations to long-term trends. This is an important study dealing with trend studies and could highlight the impact of regional policies on the dynamics of PM and O₃ concentrations. Overall, the dataset and approach are valuable, and the paper is generally well written. The work is innovative in its combined chemical characterization and trend perspective, and it is suitable for publication after minor revisions on the clarity and deeper explanation of a few aspects.

1.L135: change …. 'note' to ..' not'

Corrected. We have revised "…note only in the well-documented regions…" to "…not only…".

2. L275:  What were the primary sources of the observed Chloride?   Please identify the most plausible source for Yinchuan area as this is poorly discussed the manuscript.

We thank the reviewer for this suggestion. The chloride is predominantly emitted from primary sources, including biomass burning, coal combustion and cooking activities. However, given the negligible influence of crop residue burning in urban Yinchuan, coal combustion emerges as the most plausible dominant source of chloride, further supported by the presence of PAHs in the organic mass spectral, which are typical tracers of coal combustion.

The contribution of chloride to the total aerosol composition in Yinchuan was only 1% during summer, consistent with the substantially reduced intensity of combustion-related activities during this season.

We have incorporated the primary source of chloride in the revised manuscript,

"A similar reduction of 26.5% was observed for chloride, following the trend of organics as a response to decreased primary emissions, particularly from coal combustion."

3. L290-295: Why would higher precipitation reduce NOx significantly, yet appears to coincide with an increase in SIA (as discussed around Line 274). This seems counterintuitive without additional explanation. For example: Precipitation may reduce NOx via scavenging and decreased photochemistry, but SIA could increase if ammonia availability, aqueous-phase oxidation (for sulfate), or gas-particle partitioning (for nitrate or ammonium) is enhanced under high RH conditions.

We sincerely thank the reviewer for pointing this out. The 46% reduction in NOx from P1 to P2 is primarily attributed to decreased primary emissions rather than a direct

effect of precipitation. While wet scavenging can also remove NOx, the decreases in primary species is due to the combined influence of reduced emissions and wet removal.

However, SIA exhibit a different behavior. Despite concurrent wet scavenging, which partially offsets secondary formation, precipitation promote SIA formation through aqueous-phase processing, heterogeneous reactions and enhanced gas-particle partitioning although wet scavenging can somewhat offset the secondary production. Consequently, even NOx decreased by 46% during P2 due to decline in primary emissions, nitrate concentrations still increased by 50-100%, particularly at the nighttime, due to the heterogeneous hydrolysis of $N_2O_5$.

We have rephrased these sentences to enhance clarity and avoid ambiguity,

"Furthermore, despite lower NOx levels resulting from reduced primary emissions, nitrate concentrations in P2 were 50-100% higher than in P1, particularly at 00:00-05:00. This nocturnal enhancement was possibly due to the heterogeneous hydrolysis of $N_2O_5$ under high RH conditions, which could also account for the overall elevated nitrate levels observed in P2."

4. L287: change "It is note" to "It is noted that".

Corrected. We have revised "It is note that…" to "It is noted that…".

5. L364: revise the sentence to read better: use "exhibited strong correlation" instead of "highly correlated"

Corrected. We changed "highly correlated with" to "exhibited strong correlation with".

6. L438: The discussion on photochemical processes linking ozone formation and SOA is currently too general. A few SOA compounds were detected, but there are no explained mechanisms that connect their presence to ozone dynamics. Please expand briefly on which photochemical pathways are most relevant for $O_3$ formation in this region (e.g., VOC–NOx sensitivity regime, role of ROx chemistry, possible contribution of aromatics or biogenic VOCs... Are they any relationships of the identified SOA compounds to those pathways and if yes which ones are most relevant. Are the SOA just indicators of photochemical activity or are they direct drivers of $O_3$ changes? Please make this clear in the revised manuscript.

We appreciate the reviewer for these valuable and insightful suggestions. Ozone formation in Yinchuan is characterized by a VOC-limited regime, consistent with previous studies in urban sites. Among the various VOCs classes, alkenes and OVOCs play a disproportionately large role in driving $O_3$ production dur to their high reactivity with OH radicals. Consequently, targeted reductions in these highly reactive VOCs would be effective strategy for mitigating local $O_3$ formation during summer.

In contrast, the contribution of different VOCs groups to SOA formation follows a markedly different pattern. Aromatics including benzene, toluene, xylene, trimethylbenzene are the dominant SOA precursors in Yinchuan, contributing 51.4% to

SOA formation potential. In comparison, OVOCs such as formaldehyde and acetaldehyde contribution approximately one-third to SOA formation. Surprisingly, alkenes, despite being key drivers of $O_3$ production, contribute less than 1% to SOA mass. These results highlight a clear decoupling between the chemical species that govern $O_3$ formation and those that dominate SOA production.

Given the substantial contribution of OVOCs to $O_3$ formation, the production of ROx radicals likely originates from OVOC-related reactions, including OVOC photolysis, OVOC oxidation by OH and $NO_3$ radical. However, the detail mechanisms governing ROx production from these pathways require further careful examination. Note that our primary focus of this study is on the sources and formation of SOA; therefore, a comprehensive analysis of ROx radical chemistry is beyond the current scope and will be addressed in our future work.

We elaborated the discussion in Sec. 3.3 to enhance the readability,

"Beyond biogenic sources of SOA, we noticed substantial daytime enhancements in aromatic oxidation products, such as benzaldehyde and hydroxyisophthalic acid ($C_8H_6O_5$), likely suggesting that primary traffic-related emissions undergo further atmospheric oxidation to contribute to SOA formation. This is consistent with the oxidation of aromatics accounted for nearly half of the SOA formation potential during this campaign. Except for the traditional precursors including aromatics, alkenes and alkanes, OVOCs were found to contributed more than 30% to both ozone and SOA formation in Yinchuan, consistent with a recent study showing the critical role of OVOCs in atmospheric photochemistry (Hui et al., 2025). Given this findings, future work should prioritize elucidating the specific chemical pathways linking OVOCs to SOA and $O_3$ formation in Yinchuan, which beyond the scope of present study but are essential."